# Access to healthcare for people with disabilities in Zambia: a qualitative study

Nathaniel Scherer[1]*, Rhoda Chabaputa[2], Tamara Chansa-Kabali[3], Kofi Nseibo[4,5], Judith McKenzie[4], Martha Banda-Chalwe[6], Tracey Smythe[1,7]

1 International Centre for Evidence in Disability, London School of Hygiene & Tropical Medicine, London, United Kingdom, 2 Independent Research Consultant, Lusaka, Zambia, 3 Department of Psychology, School of Humanities and Social Sciences, University of Zambia, Lusaka, Zambia, 4 Including Disability in Education in Africa (IDEA) Research Unit, Division of Disability Studies, Department of Health and Rehabilitation Sciences, University of Cape Town, Cape Town, South Africa, 5 Department of Special Needs and Inclusive Education, Faculty of Education, Pentecost University, Accra, Ghana, 6 Centre for Research in Disability, Rehabilitation and Policy Development (CR-DRPD), Lusaka, Zambia, 7 Division of Physiotherapy, Department of Health and Rehabilitation Sciences, Stellenbosch University, Cape Town, South Africa

* nathaniel.scherer@lshtm.ac.uk

## Abstract

People with disabilities globally experience poorer health outcomes than people without disabilities. The United Nations Convention on the Rights of Persons with Disabilities emphasises that people with disabilities must have equal access to healthcare, yet evidence demonstrates barriers to access across contexts. Research on this topic is limited in Zambia, and this study therefore aimed to generate evidence on access to healthcare for people with disabilities in Zambia. In this qualitative study, we conducted in-depth interviews with 48 participants, including 16 adults with disabilities, 16 caregivers of a child with disabilities, 12 primary healthcare professionals, and four key informants from government and civil society. Participants were recruited from three districts in Lusaka Province (Lusaka, Chongwe and Kafue), the most populated province in Zambia. Participants were purposively sampled to maximise variation by sex, age, impairment type, district and locality (rural, urban, peri-urban). Data collection was completed in August 2022. Key themes were mapped against the Levesque Framework of healthcare access. Participants reported limited information on available services, stigma from community members and healthcare professionals, limited knowledge on disability and a lack of training for healthcare professionals, and challenges with inaccessible health facilities and transport. Some people with disabilities benefited from government schemes, such as the National Health Insurance Scheme, but implementation faced challenges and not all people with disabilities accessed these services. Government action is needed to improve disability-inclusive healthcare in Zambia, alleviating barriers to reduce health disparities. Recommended actions include training for healthcare professionals and improved facility accessibility.

## Author summary

In this study, we have identified a number of challenges experienced by people with disabilities in Zambia when accessing healthcare. The challenges are numerous and experienced

**Data availability statement:** Excerpts from the transcripts of this qualitative research have been made available within the article. Providing anonymised quotations in publicly-available publications was confirmed by participants during the informed consent process. Full-transcripts are not available via a public data repository. Sharing full-transcripts, even with known identifiers removed, was not approved by the ethics committees. For queries, please contact ethics@lshtm.ac.uk

**Funding:** The research in this article was supported by CBM Christoffel-Blindenmission Christian Blind Mission e.V. [grant number 4002-MYP] to TS. The funders had no role in study design, data collection and analysis, decision to publish, or preparation of the manuscript. Authors NS and RC received salary through this funding for the research undertaken.

**Competing interests:** The authors have declared that no competing interests exist.

at all stages of the healthcare journey, from identifying a need to the utilisation of services. The findings are consistent with evidence from other contexts, and provide recent evidence to support increased action for disability-inclusive healthcare in Zambia.

## Introduction

There are 1.3 billion people with disabilities globally [1]. The majority of people with disabilities (approximately 80%) live in low- and middle-income countries (LMICs). Article 25 of the United Nations Convention on the Rights of Persons with Disabilities (UNCRPD) states that "persons with disabilities have the right to the enjoyment of the highest attainable standard of health without discrimination on the basis of disability" [2]. This provision emphasises that people with disabilities should have equal access to healthcare that is free or affordable, tailored to their specific needs, and available within their communities. To support this, health professionals should receive training to ensure equitable care for people with disabilities. Additionally, States should prohibit discrimination in the provision of health insurance and prevent the discriminatory denial of healthcare.

Although most countries have ratified the UNCRPD, people with disabilities experience poorer health outcomes and higher mortality than people without disabilities [1,3,4]. Estimates indicate that all-cause mortality is 2.24 times higher in people with disabilities [3,4]. Mechanisms that contribute to health inequities include structural factors (e.g., stigma and discrimination, inequitable policies), social determinants (e.g., poverty, limited access to education) and health system challenges (e.g., inaccessible health facilities, untrained health professionals) [1]. The UNCRPD and the World Health Organization advocate for the use of data and evidence in developing health systems that improve access to healthcare and improve health outcomes for people with disabilities, especially in LMICs [1,2].

Zambia has an estimated all-age disability prevalence of 7.7% - approximately 1.5 million people [5]. The government ratified the UNCRPD in 2010, passed the Persons with Disabilities Act in 2012 (which mandates free healthcare for people with disabilities including rehabilitation and assistive technology), and developed the National Policy on Disability in 2013 [6]. The 2018 National Health Insurance Act includes exemption from mandatory contribution for people with disabilities unable to work, granting free healthcare. Additionally, people with severe disabilities qualify for the government-provided Social Cash Transfer programme, if they live in a household that meets welfare criteria, based on a Household Living Conditions Index, including household income. To be eligible, people with severe disabilities must be certified by a health practitioner. Households including a member with a severe disability are eligible to receive double the typical amount, receiving 400 Kwacha (~$18) per month [6]. To protect and uphold the right to health and the rights of persons with disabilities enshrined in law, there exists the Zambia Agency for Persons with Disabilities (ZAPD), a quasi-government institution that coordinates with Ministries and civil society to promote disability inclusion [6]. People with disabilities can obtain a disability identity card from ZAPD free of charge. This card acts as proof of disability when required under State law; for example, when accessing the Social Cash Transfer programme.

Although the rights of people with disabilities are enshrined in law and policy, a scoping review of disability research in Zambia found that people with disabilities in Zambia experience daily challenges [7]. However, the review found a dearth of recent literature on access to healthcare services for people with disabilities. Just two studies investigated access to general healthcare for people with disabilities and these were both student dissertations published over ten years ago. There is need for current evidence on healthcare access for people with

disabilities in Zambia to understand implementation of health legislation, policy and Article 25 of the UNCRPD.

Thus, the aim of this study was to explore access to healthcare services among people with disabilities in Zambia, investigating barriers and facilitators to access, as well as potential solutions to promote disability-inclusive healthcare.

## Methods

This phenomenological qualitative study utilised in-depth interviews with 48 participants, including 16 adults with disabilities, 16 caregivers of a child with disabilities, 12 primary healthcare professionals, and four key informants from government and civil society. The research has been reported in accordance with the Standards for Reporting Qualitative Research (SRQR) checklist [8].

### Study setting

The study was conducted in Lusaka Province, the most populated and densely populated province in the country. Three districts (Lusaka, Chongwe, Kafue) were selected to provide a distribution of urban, peri-urban and rural localities. Lusaka City in Lusaka district, the capital and largest city of Zambia, is largely urban. Chongwe district, approximately 40km from Lusaka, is a mix of rural and peri-urban settings. Kafue district, approximately 50km from Lusaka, is similarly a mix of rural and peri-urban settings.

Healthcare in Zambia is structured across three levels [9]. Level 1 comprises health posts, health centres, and mini-hospitals which provide community health initiatives, primary healthcare, preventative health services, and health promotion activities. Health professionals at level 1 typically include community health workers, nurses and clinical officers. Clinical officers have not completed a medical degree but are trained to diagnose and treat common illnesses and injuries. At level 2, first and second level hospitals provide secondary and curative care, such as diagnostic tests and surgeries. Level 3 includes tertiary and specialised hospitals that provide specialist care and advanced medical treatment. Level 2 and 3 are typically staffed by registered nurses and medical officers (fully qualified doctors). The levels of the health system are connected by a structured referral system. As of 2021, the country had seven specialised hospitals, seven tertiary hospitals, 36 second-level hospitals, 100 first-level hospitals, 62 mini-hospitals, 1,720 health centres and 1,388 health posts [9]. Of the 3,320 health facilities, 2,834 are Government owned, 385 are private-owned, and 101 are Faith-based facilities. In addition, traditional and indigenous medicine is provided by traditional health practitioners (also called traditional healers) within local communities. Traditional medicine includes disciplines such as herbalism and spirituality. Traditional health practitioners are organisers under the Traditional Health Practitioners' Association of Zambia.

### Participants

Participants for this study were recruited with support of organisations of persons with disabilities (OPDs), national non-governmental organisations (NGOs) and disability focal points in the community. Participants were purposively selected in order to maximise variation in line with Patton's maximum variation sampling [10]. For adults with disabilities and caregivers of a child with disabilities, participants were selected to generate variation across age, sex, type of impairment, district and locality (urban, peri-urban, rural). Impairment type was self-reported by participants and classified as physical, hearing, visual or intellectual. These categories of impairment are in line with the UNCRPD definition of disability, although we did not include psychosocial disability because of funder priorities. Hearing and visual

impairment were not grouped under the broader term of sensory impairment in order to capture the unique experiences associated with these different sensory impairments. Although we have listed visual or hearing impairment when presenting sample characteristics, we have often used the terms blind, deaf and hard of hearing in the results, based on participant preference. For health professionals and key informants, variation was sought with regards to profession, role, district and locality. Key informants included government officials working on disability, and representatives from OPDs and NGOs. To support the sampling strategy, we segmented the target group by desired criteria, aiming for equal representation across each category. From this, we sought a minimum sample of 16 adults with disabilities, 16 caregivers of a child with disabilities, and 12 primary healthcare professionals. No minimum sample size was sought for key informants, with the sample size resulting from saturation. In total, 48 participants were recruited. Table 1 presents a summary of participant characteristics.

## Data collection

Interviews were conducted between 31 May-29 November 2022. The research team was comprised of the lead author (NS), a researcher from the UK, and a qualitative researcher from Zambia (RC). RC received a three-day training from NS on disability, informed consent, research ethics, the interview guides, and considerations when interviewing people with disabilities (e.g., working with a sign language interpreter). RC provided NS with guidance for culturally appropriate research in Zambia. Four pilot interviews were conducted, in which to refine the interview guides and interview technique. These interviews were included in the final analysis.

The interview guides explored the experiences of people with disabilities (including children with disabilities and their caregivers) accessing healthcare, healthcare professionals' experience of providing care, health programmes targeting people with disabilities, and healthcare provider training on disability. Interviews focused primarily on access to general healthcare services, although questions were asked on referral and access to rehabilitation, other specialised services and traditional medicine. Interviews guides were developed by the authorship team, based on the wider literature and discussion with a Zambian OPD. They were tailored to each participant group, in order to capture the unique perspectives of patients and providers.

Interviews were conducted in Cinyanja and English. NS conducted interviews in English (his native language), whilst RC conducted interviews in Cinyanja (her native language) and English (her fluent second language). Cinyanja is the most widely spoken native language in Lusaka Province. English is the most commonly used second language nationally. Participants interviewed in their preferred language. Interviews with deaf and hard of hearing participants were conducted via a sign language interpreter. The majority of interviews were conducted in participants' homes or, in the case of health professionals and key informants, private offices at their place of work. One key informant interview was conducted remotely via online communication technology. Adult participants with an intellectual disability were asked if they wanted to be interviewed alone or with a caregiver. If they preferred or if they did not have capacity, a caregiver was interviewed as proxy. One adult with an intellectual disability interviewed alone, one interviewed alongside a caregiver, and two were interviews with a caregiver alone providing a proxy response on their relation's experience. Self-reporting by people with intellectual disabilities and communication challenges in research is important, but we also included proxy response by a caregiver so that we could investigate the experiences of people who have difficulty communicating directly, and thus may experience additional barriers to healthcare access.

Interviews lasted 30-80 minutes. At the end of each day, NS and RC discussed interview notes and emerging themes. All interviews were audio-recorded. Responses from sign language users were reported orally by the interpreter for the interviewer and audio-recording.

**Table 1. Sample characteristics.**

| Variable | Characteristic | N | % |
|---|---|---|---|
| People with disabilities | | | |
| Total | | 16 | |
| Age | 18-30 | 5 | 31% |
| | 30-60 | 6 | 38% |
| | 60+ | 5 | 31% |
| Sex | Male | 8 | 50% |
| | Female | 8 | 50% |
| Impairment type | Physical | 4 | 25% |
| | Hearing | 4 | 25% |
| | Visual | 4 | 25% |
| | Intellectual | 4 | 25% |
| Locality | Urban | 6 | 38% |
| | Peri-urban | 6 | 38% |
| | Rural | 4 | 25% |
| Interview language | Cinyanja | 7 | 44% |
| | English | 5 | 31% |
| | Sign language | 4 | 25% |
| Caregivers of children with disabilities | | | |
| Total | | 16 | |
| Child age | 0-6 | 4 | 25% |
| | 7-10 | 5 | 31% |
| | 11-14 | 7 | 44% |
| Child sex | Male | 7 | 44% |
| | Female | 9 | 56% |
| Impairment type* | Physical | 7 | 44% |
| | Hearing | 5 | 31% |
| | Visual | 4 | 25% |
| | Intellectual | 5 | 31% |
| Caregiver age | 18-30 | 1 | 6% |
| | 30-60 | 14 | 88% |
| | 60+ | 1 | 6% |
| Caregiver sex** | Male | 4 | 22% |
| | Female | 14 | 78% |
| Locality | Urban | 6 | 38% |
| | Peri-urban | 6 | 38% |
| | Rural | 4 | 25% |
| Interview language | Cinyanja | 9 | 56% |
| | English | 6 | 38% |
| | Sign language | 1 | 6% |
| Healthcare professionals | | | |
| Total | | 12 | |
| Age | 18-30 | 4 | 25% |
| | 30-60 | 8 | 75% |
| | 60+ | 0 | 0% |
| Sex | Male | 4 | 33% |
| | Female | 8 | 67% |

*(Continued)*

**Table 1.** (Continued)

| Variable | Characteristic | N | % |
|---|---|---|---|
| Role | Nurse | 7 | 58% |
| | Medical officer | 5 | 42% |
| Locality | Urban | 4 | 33% |
| | Peri-urban | 4 | 33% |
| | Rural | 4 | 33% |
| Interview language | English | 12 | 100% |
| Key informants | | | |
| Total | | 4 | |
| Age | 18-30 | 0 | 0% |
| | 30-60 | 4 | 100% |
| | 60+ | 0 | 0% |
| Sex | Male | 3 | 75% |
| | Female | 1 | 25% |
| Expertise | Healthcare | 2 | 50% |
| | Disability | 2 | 50% |
| Interview language | English | 4 | 100% |

*Participants may have a condition resulting in more than one disability/impairment type and the total may be greater than the sample size.

**In some instances, interviews were conducted with two caregivers at one time and the total is thus greater than the sample size.

Audio-recordings were transcribed verbatim. When interviews were conducted in Cinyanja, these were translated and transcribed directly into English by an independent transcriber, and checked against the audio-file by the interviewer (RC). Transcripts were anonymised and stored on a secure server.

## Data analysis

Data was analysed using thematic analysis [11]. Analysis occurred across six steps: (1) authors became familiar with the data; (2) authors developed a coding framework, with transcripts coded in NVivo 12 by NS; (3) emerging themes were identified; (4) themes were reviewed and mapped against the data; (5) themes were refined and an emerging narrative constructed; (6) participant quotes and case study narratives were extracted. During analysis, transcripts were coded by NS only, with frequent discussion and review by RC and TS, before wider discussion on themes with other co-authors. Using a single coder is recommended by Braun and Clarke (2022) in their recent guidance on thematic analysis [12].

To aid understanding, data and themes were mapped against the Levesque Framework of healthcare access (Fig 1) [13]. This framework sets out supply-side (i.e., healthcare provider) and demand-side (i.e., healthcare user) factors that contribute to access to healthcare. The framework has been used to understand access to healthcare for people with disabilities in LMICs [14]. In the results, key themes have been presented against the different stages of healthcare access outlined in the Levesque framework.

## Ethical considerations

Ethical approval was obtained from the Research Ethics Committee at the London School of Hygiene & Tropical Medicine (26568) and the University of Zambia Directorate of Research and Graduate Studies (HSSREC-2022-APR-009).

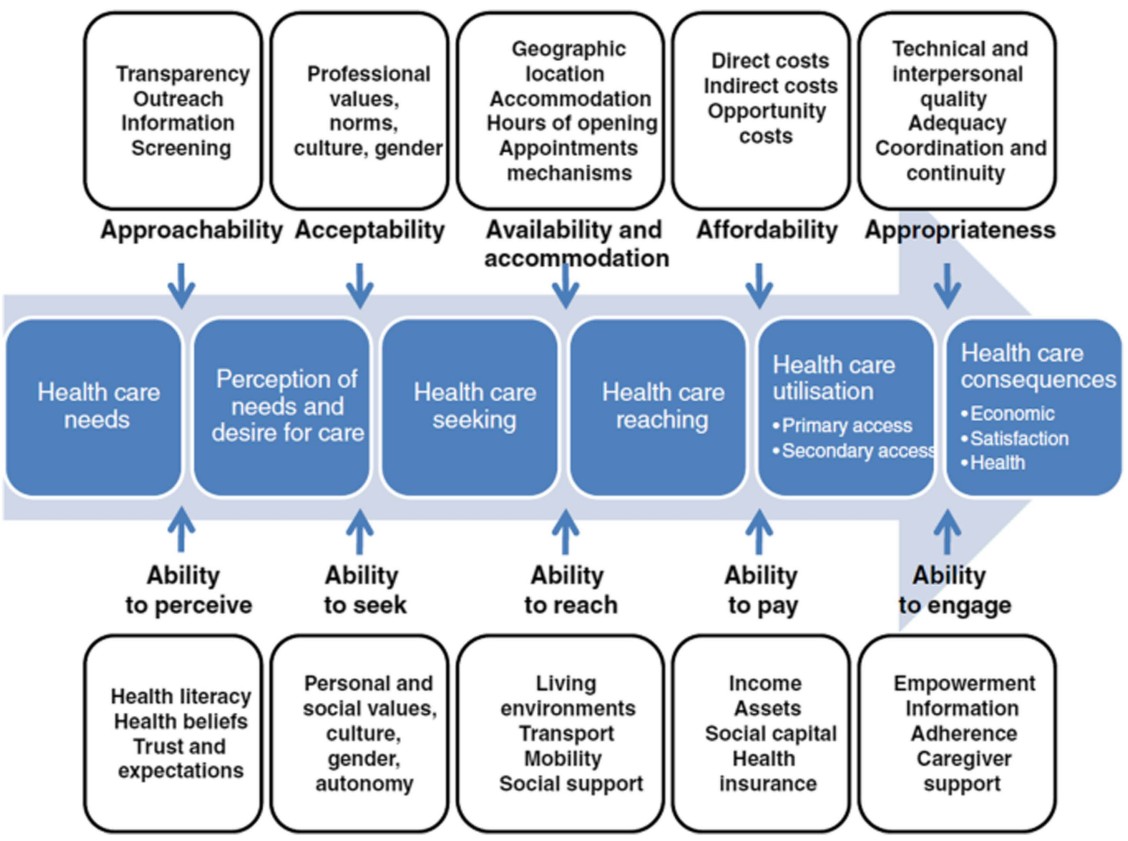

**Fig 1. Levesque Framework of healthcare access featuring supply (top) and demand-side (bottom) factors.**

Before starting each interview, researchers sought written or verbal informed consent from participants. Verbal consent was audio-recorded and a consent form was signed by a witness. People with communication or intellectual disabilities were provided a simplified information sheet, with caregiver consent and participant assent obtained, when required.

### Reflexivity of research team

Our research team comprised diverse perspectives, including disability researchers from the UK, Zambia, Zimbabwe, Ghana and South Africa. One of the researchers is a person with a disability and one is a caregiver of a child with a disability. The authors' diverse backgrounds and perspectives contributed to a comprehensive understanding of the study context. NS, the lead author, led the analysis and interpretation of findings. He does not have a disability and is from the United Kingdom, although he has worked in disability research across East and Southern Africa for a number of years. It is recognised that NS's interpretation of the findings may have been influenced by his limited experience of disability within the specific context of Zambia. However, discussion and reflection with co-authors from Zambia and other African countries, and with co-authors with lived experience of disability, enriched the analysis with diverse perspectives and mitigated potential biases.

## Results

Data highlight challenges in healthcare access for children and adults with disabilities in Zambia. Themes such as limited information, negative attitudes, transport difficulties, inaccessible

infrastructure, communication barriers, financial constraints, and the need for professional training emerge across different components of healthcare access. An overview of the themes and additional quotations are provided in S1 Table.

All findings in this section come from analysis of interviews with research participants. The authors do not present their own assumptions or additional information from the wider literature. Unless otherwise stated, the findings come from analysis across all participant groups, including perspectives from people with disabilities, caregivers, healthcare professionals and key informants. When reference is made to "participants", we refer to information consistent across participant groups. When information has come from a single group, we aim to make the appropriate reference clear (e.g., "participants with disabilities").

## Theme 1: Ability to perceive, Approachability

Ability to perceive and approachability refer to how easily individuals can identify available services, the means of accessing care and the potential benefits to their health.

**Health literacy.** Participants with disabilities and caregivers discussed two types of healthcare – general primary healthcare (e.g., under–five clinics, vaccines) and disability-specific services related to their health condition or impairment. While participants with disabilities and caregivers demonstrated general health literacy, many lacked information and knowledge on disability and disability-specific services, particularly in rural areas. Participants with disabilities and caregivers noted that primary healthcare services do not provide disability support and they do not attend these facilities when they have a concern related to disability. Despite a referral system from primary health providers to specialised services, many participants with disabilities and caregivers were not aware that these were available, leading to underutilisation of available services.

**Perception of support available.** Participants with disabilities and caregivers reported being refused support at primary health services for general health issues. Many primary health professionals saw their role for people with disabilities as referral to secondary and tertiary care only, even for minor general health concerns.

*"At the clinic they always refer, even if he's just not feeling well, like he just has... maybe he's got flu and they would say, 'No, take him to (name of hospital).' And I was thinking some of these things it was really not necessary because he didn't have like something to do with his condition. Any child, even if you have a disability, you get sick sometimes, but it's like they... I don't know whether they are scared that maybe they may do the wrong thing, so they always refer us to (name of hospital)."*

(Caregiver, female, child with physical impairment, Lusaka)

**Limited outreach targeting people with disabilities.** Regarding outreach, participants reported that many people with disabilities and caregivers were unknown to health facilities, and thus were not targeted. Although healthcare professionals and key informants recognised that there were many people with disabilities who needed care, they noted a lack of programmes to identify people with disabilities in the community, leading to a gap in outreach for both disability-specific and general healthcare needs.

*"…we need more information about us to reach the hospital, so that the hospital can be aware that [children with disabilities] are present in the community, these are the challenges they are facing and this is how we can be of help to them. You will find that they neglect you without information. This is why you find that some parents just lock their children up and*

*wait for them to die. This is why it is so important that the hospitals pay more attention to us… with that information they are able to help us."*

(Caregiver, female, child with a physical and intellectual impairment, Chongwe)

Patient registration is typically completed by community health workers. However, community health workers in Zambia are not trained to assess disability and people with disabilities often go unregistered and underserved. Some organisations have aimed to work with primary health facilities to identify people with disabilities for targeted support, but these efforts were limited. For example, in 2020, a local NGO sought support from primary healthcare in rural Chongwe to identify children with disabilities and provide disability-support, including access to assistive technology. Although the health facility assessed and registered many children with disabilities, the demand exceeded expectations and the NGO could not meet the needs of all children with disabilities. This led to disappointment and distrust in the health system among many families. Participants emphasised the importance of avoiding such situations in the future, as people with disabilities are often let down by promises that underdeliver.

**Improving information on healthcare.**  In order to increase awareness, participants suggested that information on disability, health and accessing available services be disseminated in the community through volunteers and village headmen. This has been effective in spreading information on available vaccines and COVID-19. This strategy was recommended particularly for people with disabilities who are hard-to-reach or unable to move from home easily. Other suggestions included disability 'champions'. These are people with lived experience attached to a health facility who can help sensitise people with disabilities and their families on disability, health and services available. This has been used successfully in rural and peri-urban areas for HIV and tuberculosis. Participants also recommended parent groups, similar to existing programmes, such as Safe Motherhood Champions Groups (SMAG), in which mothers with lived experience receive training to support other mothers in their local community. Participants suggested that this model could be used effectively with mothers of children with disabilities, to help them receive health information and learn simple support and rehabilitation to implement at home.

## Theme 2: Ability to seek, Acceptability

Acceptability relates to the factors that influence the appropriateness of the service available to the individual. Ability to seek refers to the individual's capacity to seek healthcare, including autonomy and individual rights.

**Negative attitudes towards people with disabilities.**  Past experiences of stigma and discrimination at primary health facilities, both from health staff and other patients, negatively impacted the acceptability of services.

*"I think their [healthcare professionals] attitude has to change. Because the way I see them, they see these children with cerebral palsy... it's like they don't see them to be human beings. Yes, they don't see them to be human beings at the end of the day."*

(Caregiver, female, child with a physical and intellectual impairment, Lusaka)

Although some participants said that health professionals were supportive, many participants with disabilities and caregivers reported negative attitudes from health professionals, which made them reluctant to seek care.

*"If I'm blind they shouldn't call me funny names… you know the others would say, 'The camera is... is not functional' [...] I've come with my personal assistant, and then instead of you*

*asking me what the problem is with me, you ask my personal assistant, 'Why have brought this man here?' […] So the language or attitude is what dehumanises a person with a disability."*

(Adult, male, visual impairment, Lusaka)

Some caregivers of children, particularly those with a severe disability, said that they rarely left the house with their child because of stigma towards disability by community members. There were calls for community sensitisation on disability to improve attitudes and increase participation in the community, including attendance at health services. Participants recommended that headmen and volunteers be trained to improve disability awareness in communities. Participants also called for structured sensitisation programmes via radio, community groups and via information in health facilities (such as posters).

**Professional values and knowledge.**  Participants noted that healthcare providers lacked knowledge on disability and supporting people with disabilities. Many people with disabilities did not attend primary healthcare because of limited knowledge among health providers. Limited knowledge on disability among health staff can also result in inappropriate medical advice and support. For example, a caregiver reported that health professionals regularly insist that their child take medication as a tablet, despite them having trouble swallowing. Participants agreed that health professionals needed more training on disability, in order to improve their quality of care and reduce negative or exclusionary attitudes. Further details on this training are provided in later sections.

As an alternative to primary healthcare, participants discussed the role of traditional healers. Many participants with disabilities and caregivers said that they did not seek support from traditional healers, as they shared false information on disability and health.

*"Traditional doctors, they cannot do anything to benefit a person with a disability because they depend on just making money. There's no help and care they can render to us […] What can I receive from a witchdoctor, what free service can I receive? Nothing."*

(Deaf adult, male, Chongwe)

Participants reported that traditional healers spread information amongst the community that disability was caused by witchcraft. As a result, community members often encouraged and pressured people with disabilities and caregivers to seek support from a traditional healer to control this witchcraft. Thus, some people with disabilities present late to modern healthcare facilities, worsening health outcomes. Participants in this study recommended that traditional healers be given training on disability, so that they no longer spread information that disability is caused by witchcraft and they promote appropriate healthcare solutions. Key informants reported that there are plans to provide disability training for members of Traditional Health Practitioners' Association of Zambia. There will also be incentives for traditional healers to support people with disabilities to access modern health services.

**Improving autonomy.**  Overall, participants said that many people with disabilities do not attend health clinics because of the challenges they face. To build trust in health facilities and to help identify the needs in the community, health professionals suggested holding sessions and half-day workshops at primary health facilities, in which to inform people with disabilities and caregivers about the services and support available, to dispel worries and to understand priority concerns of this group. These activities would require staff training and support from disability experts.

## Theme 3: Ability to Reach, Availability and Accommodation

Ability to reach, and availability and accommodation, relate to whether a health service can be physically reached, as influenced by facility accessibility and resources, individual mobility and transport, and characteristics of providers.

Throughout the interviews, participants considered a number of challenges and barriers experienced by people with disabilities when accessing and reaching healthcare facilities, including difficulties with travel, inaccessible facilities and communication challenges.

**Challenges with transport and inaccessible environments.** Participants regularly cited transportation as a significant barrier due to both availability and cost, particularly in rural and peri-urban areas where primary health facilities are often distant from people's homes. In these regions, challenging terrain, such as rocky or sandy roads, made travel difficult for those with physical or visual impairments. A participant with a visual impairment reported that he has to walk 30 minutes along rocky and uneven ground to reach the health facility, for which he requires assistance from his family members, taking them away from school and work. Even in urban settings, narrow and uneven pavements limited accessibility for people with disabilities. Across locality, caregivers of children with disabilities, especially those with a physical impairment, often had to carry them to health facilities. This becomes more difficult for older, heavier children. Caregivers said this put great physical strain on them and there were reports of musculoskeletal troubles as a result.

*"The problem I experience the most is the issue of always having to carry her on my back. Despite her being slim, she has a lot of weight and to be able to take her to the hospital, I need to carry her on my back. We do have a wheelchair, however we are unable to use it because our roads are very sandy. The effort it would take to push the wheelchair is more compared to just putting her on my back. So the wheelchair is just for home."*

(Caregiver, female, child with a physical and intellectual impairment, Chongwe)

Participants gave examples in which people with disabilities have been placed into a wheelbarrow, oxcart or similar to aid with transport to a health facility, which causes a loss of dignity and can cause physical pain. Some people borrow money from friends and family to afford transport to the clinic, but this can put strain on relationships and an individual's financial situation. Many participants with disabilities and caregivers told us that they choose not to attend a health facility because of the travel costs attached. As well as primary health facilities, this was the case for secondary and tertiary hospitals, where people with disabilities can receive disability-specific support. These are often located centrally in the district or in Lusaka, the capital, and many people do not attend these services because of the travel cost and time, and thus go unsupported.

Across the interviews, there were calls for transport support from participants with disabilities, caregivers, and healthcare professionals. Participants with disabilities asked for transport to come to them, and some families said they would be willing to pay a reasonable amount if accessible transport could reach them at home. Participants also suggested that household visits by primary health professionals become more common for people with disabilities, to relieve them of travel when it is difficult. Participants said this would be particularly beneficial for check-ups and small procedures, when handling a non-critical illness, such as a common cold. Key informants in the health district office said that it was too expensive for the government to provide accessible transport to all, although they recognised this as a priority. They called for partners, such as NGOs, to help the district health office buy vehicles to support people with disabilities with their transport needs. Key informants reported that advocacy from people with disabilities and caregivers, in-person or via social media, had an impact at national and district level and moved issues such as accessible transport higher up the priority list.

**Inaccessible facilities.** We were told that structural barriers at healthcare facilities also restricted access for many people with disabilities. Many facilities do not have ramps and other accessible design features. Even when facilities have an accessible infrastructure, they

may be lacking adaptive and accessible equipment, such as adjustable examination tables. Key informants in the district health office said that there is guidance from the government to adapt older buildings to be more accessible, however, they did not know of any examples of this in practice and said that adaptations were extremely rare.

*"One thing that came in my mind is that we have very, I think, qualified people, but the environment in which they work is not sufficiently equipped in terms of the appropriate technology that they're supposed to use. The services are there, but are they the right services? Are they appropriate? Are they modern? No."*

(Adult, male, physical impairment, Kafue)

**Communication difficulties and inaccessible health information.** Often described as a major challenge in health facilities were communication difficulties, particularly for deaf and hard of hearing patients or people with an intellectual disability. We were told that the majority of health providers did not have sign language interpretation available in health facilities. This is typically only available when a deaf person hires a sign language interpreter at their own expense or by having a hearing family member in attendance, which takes them away from employment and other responsibilities. To overcome communication difficulties, many deaf and hard of hearing patients will communicate with a health professional by writing. However, this is not suitable for people who cannot read or write. Many deaf people avoided health clinics because of difficulties communicating, which can be stressful, exhausting and makes them feel excluded.

*"I Google what is the medication for a sore throat and for a cold, then I go to the drugstore and buy. If I can't buy it, I go to the hospital, but then the doctor may see me cough and they think that I've a different illness and they give me the wrong prescription. Many Deaf people in Zambia have the same scenario, because at the hospital they don't know sign language […] Even the Government has never employed sign language interpreters in these hospitals to help Deaf people […] It is extremely hard for us to access health information. So now if you are sick you just do a self-prescription. If I'm seriously sick, yes, I cannot go to the government hospital."*

(Deaf adult, male, Lusaka)

Although sign language training for health professionals was considered worthwhile, most deaf and hard of hearing participants recognised that it was likely unfeasible to have all health personnel fluent in the national sign language. Instead, participants called for health facilities to have a sign language interpreter on staff or for government to pay for sign language interpreters to attend with a patient, instead of them having to pay out-of-pocket. People with intellectual disabilities and health professionals also experienced difficulties communicating. Health professionals asked for training on how best to work with and communicate with people with intellectual disabilities and communication difficulties. They would like training on how to understand a health concern when someone cannot verbally vocalise their symptoms. Communication challenges caused misdiagnosis and people with disabilities to avoid healthcare systems.

Further, health information was inaccessible to people with visual impairments. For example, medicines were difficult for people with a visual impairment to independently manage at home, as labels were not provided in braille. Participants called for braille on medication packets and bottles, and suggested other simple solutions, such as provision of pill boxes with identifiable raised labelling to help with medication management at home,

or simply inserting a staple into one medicine bag and not another, to help identification between two medicines.

> *"The writing is the serious challenge because I need to know what is contained in the certain medicine that I'm taking. Now I also need to have the understanding of when should I take this medication and if I want to make adjustments, how do I make adjustments, because I can't access the information which is written on the same. There is no information that is accessible…"*

(Adult with a visual impairment, male, Lusaka)

Inaccessible health information also transcends to health promotion and prevention campaigns. We were given the example of the COVID-19 pandemic, in which information on masking, social distancing, and other health guidance was not made available in accessible formats, especially for those with intellectual or sensory impairment. This increased risk and stress.

**Action to improve accessibility.** The disability identity card from ZAPD was noted as improving the experience of people with disabilities at a healthcare facility. Many health professionals provide expedited service or additional support when an individual presents this card. However, not all health professionals understood about the disability identity card. Further, there was not clear guidance on accommodations that should be given when a disability identity card is presented. Participants suggested standardised policies and procedures for health facilities to follow when they have a patient with disabilities, as well as improved training for health professionals on the disability card and reasonable accommodations to promote access. As well as health professionals, some participants with disabilities and caregivers did not know about the card and the available benefits. People with disabilities and caregivers also face difficulties accessing the disability identity card. Assessment of disability by a health professional is required before a card is issued, however, not all doctors are authorised to conduct this assessment and the majority are in Lusaka, to where travel is difficult for many. Thus, many people with disabilities do not have a disability identity card. Some NGOs and OPDs bring doctors to homes or local villages for 'assessment days' but this is not common practice. Participants requested that government provide assessment for the disability card in local communities, rather than in Lusaka.

## Theme 4: Ability to pay, Affordability

Ability to pay and affordability refer to the individual capacity to afford and spend time accessing healthcare.

**Inconsistent financial support.** We were told that many people with disabilities and caregivers receive the support they need for general health concerns, with the costs covered by the National Health Insurance Scheme and payment contribution exemptions for people with disabilities. However, participants noted that many people with disabilities did not know that they were entitled to exemption from contributions to the scheme. Also, the National Health Insurance Management Authority did not have a system to manage and monitor exemptions, although they were in discussion with the government to develop an improved system that would help ensure that eligible people with disabilities received the exemption.

**Unaffordable direct costs.** Medication was often unaffordable, especially when not available at a health facility and available only at a private pharmacy. This was of major concern to participants with disabilities and health conditions that required regular medication (such as epilespy).

*"This issue is very difficult because the medicine that they will write on the prescription for us to buy, we find that you don't even know where to start looking for the money to buy it. So what that means is that he will start having regular seizures all because we didn't have the money to buy the medication. The other challenge is that I can't leave him alone for me to go and look for the money to buy the medication. Who will I leave him with? So that is our biggest challenge."*

(Caregiver, female, child with an intellectual disability, Chongwe)

Participants said that needed medicines were rarely available at health facilities. This caused many problems, especially for people requiring regular medication. Participants called for medications to be made available first to priority patients at reduced cost, including those with disability and/or chronic illness.

Despite these concerns, affordability of primary health was generally satisfactory for participants with disabilities and caregivers, but for specialist disability support, this was not the case. In particular, assistive products were unaffordable to the vast majority, and people with disabilities often used old, ill-fitting and poorly working products. People reported using crutches that were not the correct size, hearing aids that no longer worked, and wheelchairs that were in disrepair. Often people with disabilities relied on donations and support from NGOs to receive an assistive product, but this was rare. In cases when received, people often could not afford the costs of maintenance and repair. As a result, many people with disabilities in Zambia do not use an assistive product that would support them.

## Theme 5: Ability to engage, Appropriateness

Appropriateness highlights whether the services available are suitable to address individual need, as well as the quality of service provided. Ability to engage relates to the individual's participation in decisions on their healthcare.

**Lack of training on disability for health providers.** Health professionals do not typically receive training on disability. Those interviewed often viewed disability through the medical model and had little understanding of social and rights-based models. Limited training and knowledge results in negative attitudes, exclusion of people with disabilities from primary healthcare, limited availability of disability information at primary facilities, and inappropriate referral to secondary and tertiary hospitals.

*"The challenges (to providing care), it's because we have minimal knowledge. The information which we have about disabled people, it's very minimal. I can't manage to handle a disabled person unless I refer them to the physiotherapist […] So it's a challenge, because instead of them receiving the help they need immediately, it will take maybe two to three days for them to source money to meet with the physiotherapist."*

(Clinician, female, Chongwe)

All health professionals interviewed requested training on disability. They were highly motivated to improve access to healthcare for people with disabilities, to provide information on disability, and to provide low-intensity disability-support in primary healthcare settings. Participants listed numerous topics to cover in the training, including understanding disability, disability rights, identifying disability, understanding needs, and information on the developmental stages for different conditions (e.g., cerebral palsy). Participants also wanted primary healthcare professionals to be trained on low-intensity support for caregivers of children with disabilities, including how to manage their child's disability (e.g., how best to feed a

child with feeding difficulties and how to maintain good hygiene for children with a physical impairment). Above all, participants wanted health professionals to be trained on respectful interactions that empower people with disabilities, and to be better equipped to coordinate and collaborate with people with disabilities and caregivers on individual needs and treatment approaches.

Health professionals requested that disability topics be embedded into university teaching and medical degrees, and in training for community health workers and others who do not typically attend tertiary education. Health professionals would like their training to be supplemented with national good practice guidance on disability-inclusive healthcare, which did not exist. Although extremely rare, NGOs have provided training to health professionals on disability. For example, an NGO trained health professionals in Lusaka on supporting people with intellectual disabilities. The training included interaction with people with intellectual disabilities and their families. Health professionals that attended the training were positive about its impact, feeling that it increased their confidence and knowledge. They reported that they have since supported parents and children to more regularly access health facilities.

> *"At first, I was scared […] But from the time I started training from (name of NGO), at least it gave confidence in me. I usually go to their site where they normally care for different disabilities. So that is where I gained courage and I can work with them and I can help."*

(Clinician, female, Lusaka)

**Limited coordination and implementation towards disability-inclusive healthcare.** When discussing the health system, key informants noted that Zambia has strong policies on disability inclusion, including in healthcare, but that implementation was limited, due to limited knowledge amongst government actors, limited resources, and competing priorities (such as nutrition). Government also has limited data on the number of people with disabilities and their needs, making it difficult to plan for disability-inclusive health. That said, key informants noted that there is motivation within government to promote disability inclusion and disability-inclusive health systems. For example, the Ministry of Health has developed an action plan to improve partnership between government district offices and NGOs in order to promote disability-inclusive healthcare. NGOs were viewed as able to help implement initiatives that the health office does not have the resources to deliver. However, key informants said that often international NGOs come to them with their own agenda, without collaborating with the district health office. These programmes often do not meet priority needs in local communities but instead reflect a global agenda of an international NGO. The key informants requested that NGOs conduct a needs assessment at the beginning of the planned programming in partnership with the Ministry of Health and district level officials to identify priority needs that will address local concerns. Collaboration was said to improve long-term impact and sustainability of NGO programmes.

> *"The government has the vision, the district has the vision, the direction where it wants to go, so if we can have partners who can come through to support us in some of these [health concerns] that are prevalent in the communities, I think it will go a long way… I look forward for more partners who come in this fashion, where we do the needs assessment, we discuss, and then we find a solution together."*

(Key informant in government district health office, male, Chongwe)

## Discussion

Despite motivation from the Government of Zambia to develop disability-inclusive healthcare systems, this study demonstrates that implementation is limited. People with disabilities and their caregivers in Zambia experience a number of barriers to accessing primary and specialist healthcare services, including affordability, transport, communication, and negative attitudes. In order to improve access and health management, healthcare professionals need training and guidance. Systems designed to improve disability inclusion, such as the disability identity card, require improvement to increase uptake and utilisation. Although Zambia ratified the UNCRPD in 2010, disability-inclusive healthcare needs immediate government response in line with the Convention.

The majority of the findings in this study are consistent with the wider literature on the experiences of people with disabilities when accessing healthcare globally and in Zambia, which highlights widespread barriers to healthcare access [1,14–21]. Specifically, findings under each component of the Levesque Framework are consistent with a recent meta-synthesis of qualitative research on access to healthcare among people with disabilities in LMICs, which found, as in this study, key barriers related to attitudes towards people with disabilities, inaccessible information, and practical challenges, such as transport and environmental accessibility [14]. Despite the literature on barriers to healthcare for people with disabilities increasing in recent decades, the majority is still conducted in high-income countries [19]. This study contributes to limited data in low- and middle-income countries and sub-Saharan Africa with which to inform policy and practice. Recommendations outlined by participants in this study align with the broader literature [22]. To develop a disability-inclusive health system, Zambia needs improved training for health professionals, accessible physical infrastructure, funding for support services such as interpreters, and appropriate health financing. Healthcare worker training in particular is required, in order to improve the confidence, knowledge and attitudes of healthcare professionals [23]. People with disabilities and their families also need to be empowered to be involved in strategic actions to develop an inclusive health system [1,24]. This includes roles in the planning, delivery and evaluation of policies, programmes, and interventions. This study also highlights the need for caregiver support. Caregivers are integral to child development, but as we see in this study and as consistent with other research globally and in Zambia, caregivers experience challenges accessing healthcare and receiving the information and support they need to care for their child with a disability [7,25,26]. Innovative, participatory interventions to support caregivers should be examined in the Zambian context [26,27]. Particular attention should be given to the intersectional barriers that female caregivers experience as a result of their child's disability and gender related-health inequality [28].

Government resources and financing are required to develop disability-inclusive healthcare in Zambia. Currently, the government spends 0.03% of the national budget on disability inclusion [29]. This must increase if Zambia is to develop a disability-inclusive health system. Investing in disability-inclusive health can be cost-saving in the long-run [30]. For every $1USD spent on disability-inclusive prevention, there is a $10USD return on investment [1]. There are several resources available to help effectively and cost-efficiently develop a disability-inclusive health system. For example, the Missing Billion Initiative provide a toolkit that includes guidance on system level assessment, healthcare worker training and starter kits for inclusive design [22,31]. The Initiative has also identified good practice examples to help inform global action [22]. Leveraging these resources will help the Government of Zambia achieve its disability inclusion targets.

The World Health Organization *Global report on health equity for persons with disabilities* recognises that health system factors, as outlined in this study, are not the only contributing factor to health inequities [1]. Structural factors, social determinants of health, risk factors

and health system factors interconnect to create inequalities in health between people with and without disabilities. The report advocates a multisectoral response in order for countries to deliver a disability-inclusive health system, "Addressing health inequities for persons with disabilities should not be a siloed activity conducted by the health sector in addition to other ongoing activities, but rather a strategy that is integrated into the overall efforts of a country to strengthen its health systems" [1]. Policymakers should adopt an integrated approach to promote equitable health in Zambia, addressing not only health system factors but also the structural and social determinants of health. Disability inclusion should be embedded across all sectors to reduce health disparities.

Our findings complement recent calls to action for disability inclusion in Zambia, submitted by rights-based organisations and OPDs to the United Nations Committee on the Rights of Persons with Disabilities [32–34]. In this, our findings also support the concluding observations of the Committee on the Rights of Persons with Disabilities to the initial report submitted by the Republic of Zambia [35]. In April 2024, the Committee highlighted gaps in implementation of the UNCRPD, including in adopting regulations for disability-inclusive health, the lack of awareness of rights among policymakers and health professionals, and the lack of training for medical professionals on the rights of people with disabilities. The Committee further notes the need to develop community awareness-raising campaigns, enhance the capacity and resources of ZAPD to implement its mandate, to strengthen disability focal points, and to improve national disability data. Throughout, the Committee recommends that people with disabilities, their families, and their representative organisations be involved in the planning and execution of disability-inclusive policies and strategies.

## Strengths and limitations

This study has several strengths. The sample was diverse, gathering a number of perspectives from rural, peri-urban and urban localities across three districts in Lusaka Province. Participants represented a range of disabilities, healthcare professions and national disability representatives. The varied sample, reflexivity and review of wider literature allowed triangulation of findings to strengthen interpretation. Utilising the Levesque framework to aid analysis helped assess the complex process of healthcare access and contributes to the wider literature using this framework with different populations [36].

With regards to limitations of the study, we must recognise that the research was conducted in Lusaka Province only. We aimed to sample participants from urban, peri-urban and rural areas across three districts, but we must acknowledge that findings may not reflect the situation across the country. This is a recurrent issue across disability research in Zambia [7]. It is important that additional research be conducted in different regions of the country to inform disability-inclusive healthcare. Next, we acknowledge that the coding of transcripts was completed by one author only. Some argue that dual coding may improve reliability in thematic analysis, but in their most up-to-date guidance, Braun and Clarke (2022) do not recommend multiple coders as a way to guarantee accurate analysis, and instead recommend using a single coder to promote researcher subjectivity and interpretation as a valued resource in non-positivist thematic analysis [12]. Although multiple coders were not used, our authorship team regularly discussed the coding and emerging themes in order to promote richer insights into the data. Thirdly, it is worth noting the higher number of female caregivers interviewed, compared to male. In Zambia, female caregivers are typically the primary caregiver of children with disabilities, and during recruitment we were often asked to speak to the female caregiver, even if the male caregiver was available. The challenges that male caregivers experienced may therefore be underrepresented in the findings and further research with this group may be beneficial, especially as research indicates that positive father-child interactions

are associated with positive social, emotional and cognitive outcomes in children [37,38]. Finally, caregivers provided proxy response for two participants with intellectual disabilities, which limited self-report by people with intellectual disabilities themselves. As we recognise in the methods, self-report is important and proxy response may result in bias, as information is not coming from the person directly. There is the risk that the information would not always align with the experiences of the individual. However, we believed it was important to capture the experiences of people with difficulties communicating, as they may face unique barriers to healthcare access. We aimed to mitigate bias by speaking with a proxy respondent who provides primary care and is present with the individual when seeking healthcare. We guided the proxy respondent to reflect on the experiences of the individual, rather than themselves.

## Conclusion

People with disabilities in Zambia experience a number of challenges to healthcare access. Government action is needed to develop a disability-inclusive health system, including improved training for healthcare providers, improved accessibility at facilities, and improved attitudes of community members and healthcare professionals. Action should be taken with people with disabilities and their representative organisations at the forefront of planning, delivery, monitoring and evaluation.

## Supporting information

**S1 Table. Overview of key themes and additional quotes.**
(DOCX)

**S1 Appendix. PLOS 'Inclusivity in global research' form.**
(DOCX)

## Acknowledgements

We would like to thank all of the participants who generously gave their time to take part in this study. Thank you to CBM Zambia, Leonard Cheshire Zambia, the University of Zambia, and the Zambia Federation of Disability Organisations for supporting us to identify participants. Thank you to Joerg Weber, Sarah Rule and to everyone at CBM Zambia for supporting study setup and logistics. Thank you to Marjolein Meande-Baltussen for her insight to aid interpretation of findings.

## Author contributions

**Conceptualization:** Nathaniel Scherer, Rhoda Chabaputa, Tamara Chansa-Kabali, Kofi Nseibo, Judith McKenzie, Martha Banda-Chalwe, Tracey Smythe.

**Formal analysis:** Nathaniel Scherer, Rhoda Chabaputa, Tamara Chansa-Kabali, Kofi Nseibo, Judith McKenzie, Martha Banda-Chalwe, Tracey Smythe.

**Funding acquisition:** Nathaniel Scherer, Tracey Smythe.

**Investigation:** Nathaniel Scherer, Rhoda Chabaputa, Tracey Smythe.

**Methodology:** Nathaniel Scherer, Rhoda Chabaputa, Tracey Smythe.

**Project administration:** Nathaniel Scherer, Rhoda Chabaputa, Tamara Chansa-Kabali, Tracey Smythe.

**Resources:** Tamara Chansa-Kabali, Martha Banda-Chalwe.

**Supervision:** Tracey Smythe.

**Visualization:** Nathaniel Scherer, Rhoda Chabaputa, Tamara Chansa-Kabali, Kofi Nseibo, Judith McKenzie, Martha Banda-Chalwe, Tracey Smythe.

**Writing – original draft:** Nathaniel Scherer.

**Writing – review & editing:** Rhoda Chabaputa, Tamara Chansa-Kabali, Kofi Nseibo, Judith McKenzie, Martha Banda-Chalwe, Tracey Smythe.

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
