## [Decision Letter · Decision Letter 0]

4 Dec 2024

PGPH-D-24-02425

Access to healthcare for people with disabilities in Zambia: a qualitative study

Dear Dr. Scherer,

Thank you for submitting your manuscript to PLOS Global Public Health. After careful consideration, we feel that it has merit but does not fully meet PLOS Global Public Health’s publication criteria as it currently stands. Therefore, we invite you to submit a revised version of the manuscript that addresses the points raised during the review process.

We look forward to receiving your revised manuscript.

Kind regards,

Annesha Sil, Ph.D.

Staff Editor

Journal Requirements:

2. In this instance it seems there may be acceptable restrictions in place that prevent the public sharing of your minimal data. However, in line with our goal of ensuring long-term data availability to all interested researchers, PLOS’ Data Policy states that authors cannot be the sole named individuals responsible for ensuring data access (http://journals.plos.org/plosone/s/data-availability#loc-acceptable-data-sharing-methods).

3. Please provide separate figure files in .tif or .eps format.

4. Please provide an Author Summary. This should appear in your manuscript between the Abstract (if applicable) and the Introduction, and should be 150–200 words long. The aim should be to make your findings accessible to a wide audience that includes both scientists and non-scientists. Sample summaries can be found on our website under Submission Guidelines: 

https://journals.plos.org/globalpublichealth/s/submission-guidelines#loc-parts-of-a-submission

5. We have noticed that you have uploaded Supporting Information files, but you have not included a list of legends. Please add a full list of legends for your Supporting Information files after the references list. 

Additional Editor Comments (if provided):

Reviewers' comments:

Reviewer's Responses to Questions

**Comments to the Author**

1. Does this manuscript meet PLOS Global Public Health’s publication criteria ? Is the manuscript technically sound, and do the data support the conclusions? The manuscript must describe methodologically and ethically rigorous research with conclusions that are appropriately drawn based on the data presented.

Reviewer #1: Yes

Reviewer #2: Yes

2. Has the statistical analysis been performed appropriately and rigorously?

Reviewer #1: N/A

Reviewer #2: N/A

3. Have the authors made all data underlying the findings in their manuscript fully available (please refer to the Data Availability Statement at the start of the manuscript PDF file)?

Reviewer #1: Yes

Reviewer #2: Yes

4. Is the manuscript presented in an intelligible fashion and written in standard English?

Reviewer #1: Yes

Reviewer #2: Yes

5. Review Comments to the Author

Reviewer #1: I really enjoyed reading this article. Highlighting the perspectives of individuals with disabilities is crucial, as their voices are often marginalized even in high-income countries (HICs) with greater resources. Providing a platform for these voices adds significant value to both the research and the broader conversation on inclusion and accessibility.

Regarding the feedback format, I’ll offer a general comment on each section, followed by specific suggestions aimed at enhancing the quality and clarity of the work.

Abstract

• The abstract is well-structured and provides a comprehensive overview of the study. I recommend removing the last sentence in the background section, ‘This study employs qualitive method to generate new evidence’, as this content belongs more appropriately in the methods section. Instead, this part could be rephrased as: ‘In this qualitative study, we conducted in-depth interviews with 48 participants…”.

Introduction

• The introduction provides valuable context on the topic and is well-proportioned in length. However, the phase ‘than people with disabilities’ appears repeatedly throughout the section and may lead to redundancy. For better fluency, I suggest removing this phrase on line e58 when discussing all-cause mortality.

• Additionally, the introduction could benefit from a smoother transition between the second and third paragraphs. Adding 'especially in LMICs' to the end of the sentence concluding with 'and improve health outcomes for people with disabilities' on line 63 may strengthen this connection.

• Currently, the connection is made with the study’s objective, but it might be more effective to place the objective at the end of the introduction where it naturally aligns with lines 88 and 89. I suggest revising the objective to clearly encompass the study’s focus on barriers and facilitators to access, as well as potential solutions, as discussed in the results.

Method

• The methodology is well-explained and, while qualitative research may not follow strict standardization, it provides sufficient detail for replicability.

Study setting

• The classification of location is somewhat overlapping here, as it differentiates people by both district within Lusaka province and location type. From what I understand, the three districts were chosen to represent urban, peri-urban, and rural locations; however, only Lusaka is entirely urban, while Chongwe and Kafue are mixed. To improve clarity, I suggest acknowledging the participants' districts of origin but using location type as the sole classification and removing the district category from the table.

• Additionally, consider standardizing terminology throughout the text, using either 'location' or 'locality' consistently.

The section also includes a description of Zambia’s healthcare system structure. While relevant, this information may fit better in the introduction, especially since all health professional participants are from primary healthcare settings. Summarizing it briefly in the introduction will better set the context.

Participants

• It may be helpful to explain the rationale behind the classification of disability used in the study. For instance, why was there a differentiation between visual and hearing disabilities, rather than grouping these as sensory disabilities? A brief explanation could help readers understand this choice.

Data collection

• For clarity, please ensure a logical order of information. For example, details on the language used in interviews (lines 144–148) would be more coherent in the third paragraph (around line 160), as it relates more closely to procedural specifics.

There appears to be an important limitation concerning adult participants with intellectual disabilities. Using caregivers as proxies may not provide accurate insights; literature indicates that caregivers' reports on health experiences do not always align with the actual experiences of individuals themselves. For an example of this topic, please see https://jamanetwork.com/journals/jamapediatrics/fullarticle/2769779

When direct communication with participants was not possible, this represents a barrier that should be recognized as a limitation. If participants' intellectual disabilities prevented complex communication even with augmentative speech strategies, the role of caregivers as informants might be acknowledged as a separate participant category, such as 'Caregivers of Adults with Disabilities.' Including this as a limitation could strengthen the study's transparency.

Data analysis

• The decision to use the Levesque Framework for healthcare access is well-suited to the study's objectives. Along with the figure, consider mentioning the framework's main components and key themes within each part to give readers a clearer understanding of its application in this context.

Reflexivity

• I appreciate the inclusion of this section, as it shows the authors’ commitment to addressing potential biases. It may be more appropriately placed immediately after 'Ethical Considerations,' and I suggest revising the title to more precisely reflect the content.

Results

• The results presented in this paper are highly engaging, although I have two main concerns. First, the term 'participant' is used ambiguously throughout. Given the various participant categories, it would be helpful to clarify each instance to indicate whether it refers to all participants or specific categories. Authors have done it in most cases but others remain ambiguous.

I also suggest refining the language of the results section to clearly convey that this information reflects the study’s findings. Using verbs such as 'reported,' 'noted,' 'mentioned,' or 'stated' may enhance readability and provide consistency in reporting participants' perspectives. Frequently, it is confusing whether the statement comes from the results or it may be an assumption.

To improve clarity for readers, consider renaming and numbering the themes, e.g., 'Theme 1: Ability to Perceive and Approachability.' Additionally, briefly specify Levesque’s categorization within each theme (e.g., supply-side vs. demand-side). Within each theme, short subtitles could be added at the start of paragraphs, formatted in italics, to guide readers through key points. Here’s an example reformulation for theme 1 following this structure:

“Theme 1: Ability to Perceive and approachability”

General health literacy. People with disabilities and caregivers discussed two types of healthcare – general primary healthcare (e.g. under-five, vaccines) and disability-specific services [deleted redundant info]. [Deleted ‘while’] Participants reported general health literacy and many lacked information and knowledge on disability, disability-specific services and referral system, especially in rural areas. They [participants and caregivers] also noted that primary healthcare services do not provide disability support and they do not attend these facilities when they have a concern related to disability [deleted information about referrals, noted before], leading to underutilization of available services.

Additionally, people with disabilities reported being refused support at primary health services for general health issues and, accordingly, many primary health professionals reported seeing their role for people with disabilities as referral to secondary and tertiary care only. [Deleted information about transport because it belongs to another theme].

(quote)

Systems outreach. Health professionals….”

Ability to Perceive, approachability

• The information beginning on line 253 with 'For example' seems more relevant to the discussion section. For instance, the phrase 'some organizations have aimed to work with primary health facilities' does not appear to be directly derived from participants' responses. Clarifying whether this information is part of the study’s findings or general context would be helpful.

• Similarly, the third paragraph of this theme would benefit from a focus on participants’ suggestions regarding solutions to the two barriers highlighted earlier in this theme. Additional context could be reserved for the discussion.

Ability to Seek, acceptability

• On line 277, it would be helpful to clarify if participants directly reported the negative impact discussed.

• The statement 'Participants recommended that headmen and volunteers be trained to improve disability awareness in communities' appears to overlap with content in the previous theme and may not be needed here.

• Consider including a quote after the first paragraph of the theme to strengthen the participants' perspectives. Then, instead of explaining the quote (lines 292–294), a summary, such as 'addressing the patient directly rather than talking to an assistant,' might add clarity.

• Paragraph 7 (line 304) and its associated quote repeat information from the previous theme on training, which may lead to redundancy.

Ability to Reach, Availability and Accommodation

• The second paragraph could be restructured for clarity, focusing on the information flow. Here’s a suggested revision: 'Participants regularly cited transportation as a significant barrier due to both availability and cost, particularly in rural and peri-urban areas where primary health facilities are often distant from people’s homes. In these regions, challenging terrain, such as rocky or sandy roads, made travel difficult for those with physical or visual impairments. Even in urban settings, narrow and uneven pavements limited accessibility for people with disabilities.

• After this paragraph, consider grouping insights by topic: first, discuss transportation costs, then accessibility challenges, and finally, explore solutions, including those appropriately reported as loss of dignity or challenging (e.g. asking for money).

• In paragraph 8 (line 390), results are presented in the present tense rather than the past, which may disrupt consistency. Also, rather than a detailed example, consider a summarized version to contextualize the quote more effectively.

• In paragraph 11 (line 422), 'Participants called for Braille on medication packets and bottles,' would read more clearly with the addition of 'and suggested other simple solutions, such as…

• On line 434, confirm if ‘This increased risk and stress’ is participant-reported.

• Lines 439-440 contain repetitive language: ‘There were calls’ appears twice and could be changed for fluency.

• To avoid redundancy, consider removing detailed descriptions of disability assessment procedures (lines 444-448).

• In the final paragraph of this theme, ‘people’ is mentioned but it’s unclear who this refers to —clarifying this term would improve understanding.

Ability to pay, affordability

• In the second paragraph, delete 'In addition to issues with transport cost, discussed above,' as it’s close enough to refer back without additional prompting.

• Subheadings within this theme, such as 'Social Support,' 'Medication Affordability,' and 'Service Affordability,' may also help guide the reader through key points.

Ability to engage, appropriateness

• On line 484, consider whether the 'medical or social view of disability' is a reported outcome from participants or a conclusion drawn from results.

• At the end of page 20 (line 492), add an 's' to the end of 'condition' to maintain grammatical accuracy.

• In the third paragraph of this theme, the example of training by NGOs would be more suitable in the discussion.

• In the subsequent paragraph, clarify if the information about the government is reported by participants or drawn from literature (if the latter, it should be included in the discussion instead).

Discussion

• The discussion could start with a concise summary of findings that directly addresses the study’s objectives. Consider restructuring to better highlight where this study’s findings align or diverge from previous literature. Currently, literature alignment is acknowledged but not explained, so expanding on this could enrich the discussion. Additionally, incorporating the points identified for discussion during the results presentation would provide valuable context here.

Strengths and limitations

• This section is well-articulated, with clearly acknowledged limitations. I recommend including the limitation regarding caregiver proxies as noted earlier in this review. Additionally, the final statement would benefit from a more nuanced reflection on gender roles: although women are recognized as the primary caregivers, this largely reflects a broader, patriarchal social structure rather than a unique factor within this context.

• The observed underrepresentation of male caregivers may be linked to limited paternal involvement in child-rearing, which has implications for children with disabilities. Research in high-income countries (HICs) suggests that when fathers are less involved, it can negatively affect early child development—a critical stage for children with disabilities, where many intervention programs are focused.

• In terms of access, this gender perspective reveals how barriers faced by women in accessing healthcare may indirectly restrict children’s access as well. Women, who often advocate for their children in healthcare settings, may find their voices overlooked or mistreated, potentially hindering timely and appropriate care for children with disabilities.

Reviewer #2: The authors conducted a qualitative research study in which they performed in-depth interviews about healthcare access for people with disability with 48 participants, including 16 adults with disabilities, 16 caregivers of children with disabilities, 12 primary healthcare professionals, and four key informants from government and civil society. Participants were recruited from three districts in Lusaka Province: Lusaka, Chongwe, and Kafue. Key themes were mapped against the Levesque framework of healthcare access, and the authors presented various key findings regarding the difficulties faced by this population in that specific area of Zambia.

This work presents novel and valuable findings and offers insights into how institutions in Zambia can effectively approach and enhance healthcare systems to improve accessibility, approachability, and acceptability, among other factors. It is well written and contributes significantly to the field. I commend the authors for their demonstration of cultural humility in the reflexivity section of the paper, where they emphasize how a diverse group of authors can contribute to the advancement of the academic field, and I was also impressed by their acknowledgment of the challenges in their study, particularly regarding the lead author's background, along with their efforts to address these issues. Below are a few comments listed.

Comments

• The sample characteristics in Table 1 could include the primary language spoken or the language in which the interviews were conducted. It would also be beneficial to add the ages of caregivers, healthcare professionals, and key informants.

• Could you expand on the process of developing the interview guides and indicate whether there were differences among the guides used for different groups or languages?

• It is stated that the interviews were conducted in English, Cinyanja, or Icibemba. Could you explain and provide data on how many interviews were conducted in each language and describe the process for determining when interviews would be conducted in English?

• In the methods section, it is mentioned that “ …authors developed a coding framework, with transcripts coded in NVivo 12 by NS.” Could you clarify whether NS was the only coder? Additionally, could you specify whether an intercoder reliability process was followed, and if so, provide information regarding this?

• The methods section states that “two adults with disabilities were interviewed with a caregiver providing a proxy response.” Could you please explain the interview process involving proxy responses in more detail?

• It would be beneficial to present the sex of the respondents along with the quotes.

• Lastly, the paper would benefit from increased diversity and quantity in the quotes used, as not all ideas discussed in the text are supported by the evidence presented, either in the main text or the supplementary material, such as experiences with the disability identity card from ZAPD or the role of traditional healers. Furthermore, almost half of the quotes come from caregivers.

6. PLOS authors have the option to publish the peer review history of their article (what does this mean? ). If published, this will include your full peer review and any attached files.

**Do you want your identity to be public for this peer review?** For information about this choice, including consent withdrawal, please see our Privacy Policy .

Reviewer #1: No

Reviewer #2: No

---

## [Decision Letter · Decision Letter 1]

17 Jan 2025

PGPH-D-24-02425R1

Access to healthcare for people with disabilities in Zambia: a qualitative study

Dear Dr. Scherer,

Thank you for submitting your manuscript to PLOS Global Public Health. After careful consideration, we feel that it has merit but does not fully meet PLOS Global Public Health’s publication criteria as it currently stands. Therefore, we invite you to submit a revised version of the manuscript that addresses the points raised during the review process.

Please respond to Reviewer 2's comments regarding the use of a single coder, and any literature to support this approach in your study design.

We look forward to receiving your revised manuscript.

Kind regards,

Jennifer Tucker, PhD

Staff Editor

Journal Requirements:

Additional Editor Comments (if provided):

Reviewers' comments:

Reviewer's Responses to Questions

**Comments to the Author**

1. If the authors have adequately addressed your comments raised in a previous round of review and you feel that this manuscript is now acceptable for publication, you may indicate that here to bypass the “Comments to the Author” section, enter your conflict of interest statement in the “Confidential to Editor” section, and submit your "Accept" recommendation.

Reviewer #1: All comments have been addressed

Reviewer #2: All comments have been addressed

2. Does this manuscript meet PLOS Global Public Health’s publication criteria ? Is the manuscript technically sound, and do the data support the conclusions? The manuscript must describe methodologically and ethically rigorous research with conclusions that are appropriately drawn based on the data presented.

Reviewer #1: Yes

Reviewer #2: Yes

3. Has the statistical analysis been performed appropriately and rigorously?

Reviewer #1: N/A

Reviewer #2: N/A

4. Have the authors made all data underlying the findings in their manuscript fully available (please refer to the Data Availability Statement at the start of the manuscript PDF file)?

Reviewer #1: Yes

Reviewer #2: Yes

5. Is the manuscript presented in an intelligible fashion and written in standard English?

Reviewer #1: Yes

Reviewer #2: Yes

6. Review Comments to the Author

Reviewer #1: I appreciate the authors' efforts in addressing the comments provided. While they chose not to implement certain suggestions, the manuscript is now well-structured and offers broader and more comprehensive conclusions. Congratulations on this valuable work and for amplifying the voices of people with disabilities—an essential contribution to public health research and advocacy.

Reviewer #2: I appreciate the authors' efforts in incorporating the feedback into their work. This paper has many strengths and effectively highlights the voices and issues crucial to people with disabilities in Zambia. The importance of conducting this work cannot be overstated, as it sheds light on often-overlooked perspectives and challenges.

My main concern, however, is the lack of a second coder, which could affect the reliability of the findings. While the authors regularly discussed the coding process, which may be sufficient to ensure reliability, I wonder if you could find additional support in the literature for the use of a single coder.

Overall, this paper makes a valuable contribution to the field, advancing our understanding of disability-related issues in Zambia and offering important insights for future research and policy development.

7. PLOS authors have the option to publish the peer review history of their article (what does this mean? ). If published, this will include your full peer review and any attached files.

**Do you want your identity to be public for this peer review?** For information about this choice, including consent withdrawal, please see our Privacy Policy .

Reviewer #1: No

Reviewer #2: No

---

## [Decision Letter · Decision Letter 2]

18 Feb 2025

Access to healthcare for people with disabilities in Zambia: a qualitative study

PGPH-D-24-02425R2

Dear Dr Scherer,

We are pleased to inform you that your manuscript 'Access to healthcare for people with disabilities in Zambia: a qualitative study' has been provisionally accepted for publication in PLOS Global Public Health.

Best regards,

Julia Robinson

Executive Editor

Reviewer Comments (if any, and for reference):

Reviewer's Responses to Questions

**Comments to the Author**

1. If the authors have adequately addressed your comments raised in a previous round of review and you feel that this manuscript is now acceptable for publication, you may indicate that here to bypass the “Comments to the Author” section, enter your conflict of interest statement in the “Confidential to Editor” section, and submit your "Accept" recommendation.

Reviewer #1: All comments have been addressed

Reviewer #2: All comments have been addressed

2. Does this manuscript meet PLOS Global Public Health’s publication criteria ? Is the manuscript technically sound, and do the data support the conclusions? The manuscript must describe methodologically and ethically rigorous research with conclusions that are appropriately drawn based on the data presented.

Reviewer #1: Yes

Reviewer #2: Yes

3. Has the statistical analysis been performed appropriately and rigorously?

Reviewer #1: N/A

Reviewer #2: N/A

4. Have the authors made all data underlying the findings in their manuscript fully available (please refer to the Data Availability Statement at the start of the manuscript PDF file)?

Reviewer #1: Yes

Reviewer #2: Yes

5. Is the manuscript presented in an intelligible fashion and written in standard English?

Reviewer #1: Yes

Reviewer #2: Yes

6. Review Comments to the Author

Reviewer #1: I think the article is ready for publication.

Reviewer #2: I appreciate the authors' thorough efforts in addressing the provided comments. The manuscript now presents clearer and more comprehensive conclusions. Congratulations on this important contribution to public health research and advocacy, particularly in amplifying the voices of people with disabilities.

7. PLOS authors have the option to publish the peer review history of their article (what does this mean? ). If published, this will include your full peer review and any attached files.

**Do you want your identity to be public for this peer review?** For information about this choice, including consent withdrawal, please see our Privacy Policy .

Reviewer #1: No

Reviewer #2: No
